# Comparing the Efficiency of Valved Trocar Cannulas for Pars Plana Vitrectomy

**DOI:** 10.3390/bioengineering12040431

**Published:** 2025-04-19

**Authors:** Tommaso Rossi, Giorgio Querzoli, Giov Battista Angelini, Camilla Pellizzaro, Veronica Santoro, Giulia Rosari, Mariacristina Parravano, David H. Steel, Mario R. Romano

**Affiliations:** 1IRCCS—Fondazione Bietti ONLUS, 00184 Roma, Italy; g.angelini123@gmail.com (G.B.A.); camilla.pellizzaro@fondazionebietti.it (C.P.); veronica.santoro@santannapisa.it (V.S.); giulia.rosari@fondazionebietti.it (G.R.); mariacristina.parravano@fondazionebietti.it (M.P.); 2DICAAR, Università degli Studi di Cagliari, 09123 Cagliari, Italy; 3Newcastle University, Newcastle NE1 7RU, UK; david.steel@newcastle.ac.uk; 4Department of Biomedical Science, Humanitas University, 20072 Milan, Italy; mario.romano.md@gmail.com

**Keywords:** pars plana vitrectomy, flow rate, head loss, valved trocar cannulas

## Abstract

Purpose: To compare the efficiency of different manufacturers’ valved cannulas (23G, 25G and 27G) (Alcon, Bausch & Lomb, BVI, DORC, Optikon) in maintaining intraocular pressure during vitrectomy by measuring leak pressure and the difference between set and actual intraocular pressure, under BSS and air infusion. Methods: A BSS-filled reservoir was connected to a model eye allowing placement of leak-proof valved cannulas. A pressure sensor was interposed and the bottle height increased until leakage occurred. Air leakage was measured by connecting an air pump to different manufacturers’ valved cannulas, inserted upside down to blow air against the valve with inside-out direction and immersed in soapy water to detect air bubbling. Results: The average BSS leaking pressure was 7.69 ± 0.77 mmHg for 23G, 9.92 ± 0.57 mmHg for 25G and 7.57 ± 0.80 mmHg for 27G. The 25G valved cannulas opened at higher pressure (*p* < 0.05). The difference between set and actual pressure in BSS never exceeded 4 mmHg. Leakage pressure under air ranged between 10 and 55 mmHg. The 27G valves opened at an average 47.2 ± 3.9 mmHg vs. 29.4 ± 7.2 for 25G and 24.1 ± 16.5 for 23G (27G vs. other gauges *p* < 0.05). The difference between set and actual pressure under air infusion never exceeded 2 mmHg. Conclusion: Despite significant differences, all tested valved cannulas satisfy safety criteria by keeping a surgically negligible difference between the set and actual intraocular pressure. The minimal leakage measure may act as a safety pressure damper under critical conditions.

## 1. Introduction

Pars Plana Vitrectomy (PPV) is an effective surgical procedure for several vitreoretinal diseases including complex retinal detachment, proliferative diabetic retinopathy, macular surgery and vitreous hemorrhage. Regardless of surgical indication, a safe removal of the vitreous gel requires invariant pressure and volume of the vitreous chamber [1] throughout the procedure. The stability of volume and pressure, in fact, reduces the incidence of dreaded complications such as bleeding [2], choroidal detachment [3] and vitreous and retinal incarceration [2].

The introduction of transconjunctival PPV in 2002 [4], and further improvements including small gauge surgery [5] and high-speed vitrectomy [6], led to the introduction of sclerostomy trocar cannula systems [7] as a safe and practical entry site, however they exposed the eye to the risks of severe, intermittent hypotony as instruments were removed and the trocar lumen was left open. Sclerostomy trocar cannulas, in fact, improve the safety of entry sites and are almost leak-proof when the instruments are inserted within the cannula, due to minimal dimensional tolerance [8], but they instantaneously turn into high-flow egress sites upon the instruments’ removal. The introduction of valved cannulas represented an elegant and valid solution to this problem [9]. Previous studies on valved cannulas demonstrated their efficacy [10] and safety [9], but also revealed drawbacks related to the difficulty of introducing diamond-dusted scrapers through them [11] and the risk of heavy liquid migration into the anterior chamber [12].

Valved cannulas’ efficiency and design not only determine the intraocular pressure but create friction with intraocular instruments’ shafts, largely impacting the surgeon’s performance. There are very few papers comparing the efficiency of valved cannulas in terms of intraocular pressure stability while infusing either Balanced Salt Solution (BSS) or air.

The purpose of the present paper is to present a comparison of the efficiency of leading brands’ valved trocar cannulas of all gauges, including the pressure at which they start leaking and the amount of pressure drop they cause during surgery, under both air and BSS infusion.

## 2. Materials and Methods

### 2.1. Experimental Setting

Valved trocar cannulas (23G, 25G and 27G) from Alcon (Fort Worth, TX, USA), Bausch & Lomb (Laval, Canada), BVI Medical (Waltham MA, USA), DORC (Zuidland, The Netherlands) and Optikon 2000 (Roma, Italy) were tested (Figure 1).

We investigated the behavior of trocar valves in response to the pressure exerted by Balanced Salt Solution (BSS) and air. In each setup, the pressures were measured by a relative pressure probe (Keller, PD-33X, Switzerland) connected to a National Instrument board (USB 6211), with a data acquisition (DAQ) interface. A mock vitreous chamber made of Plexiglas was used for the simulation of the surgical configuration. The internal volume of the chamber was spherical and closed at the top by a rubber membrane sealed by a stainless-steel ring fixed at the Plexiglas body with screws. The volume of the mock vitreous chamber was 4 mL and the distance between the rubber membrane and the opposing wall was 18 mm to reproduce the width of the human vitreous chamber (Figure 2).

### 2.2. BSS Infusion Setup

The BSS infusion system consisted of a reservoir placed at a known piezometric height, frequently refilled to maintain a constant BSS level, and 4 mm tubing that conveyed the fluid to a 3-way stopcock connecting the infusion line to the mock vitreous chamber (Figure 3A).

To assess the drop in pressure as a function of the flow (i.e., trocar leakage due to relative valve incontinence), two valved trocar cannulas were inserted into the rubber membrane of the mock vitreous chamber using the manufacturer’s trocar blade, and were then removed. The tests were performed by varying the height of the water level of the infusion tank between 0.10 m and 0.80 m (7–58 mmHg) higher than the mock vitreous chamber (24 height levels indicated with Z in Figure 3A), with the infusion fed by gravity. The range considered critical (between 0.10 and 0.27 m or 7–20 mmHg) was investigated by varying the elevation in 10 mm steps (0.7 mmHg). Above this level, the steps were incremented by 70 mm (5 mmHg), up to the highest elevation of 0.80 m, to investigate the behavior at higher pressures.

In a second series of experiments, valve-opening pressure (i.e., the pressure at which spontaneous leakage occurred) was assessed by inserting a single trocar cannula into the rubber membrane of the mock vitreous chamber (Figure 3B). The valve-opening pressure was taken as the pressure measured by the pressure probe when a leak was visibly appreciable.

A reference pressure curve in the absence of any leak was obtained by using a sealed mock chamber membrane and increasing the piezometric height of the BSS tank. These measures were compared to the results obtained with trocar insertion.

### 2.3. Air Infusion Setup

For the measurements in air, we used the air pump of the R-Evolution CR800 combined phacoemulsification and vitrectomy console (BVI, Whaltam, MA, USA) and connected the circuit described below that, in this case, was filled with air. To measure the opening pressure of the valves, the setup in Figure 4 was used: A 3-way tap allowed the connection of the infusion line, the pressure probe, and the tested cannula. The valved trocar cannula being tested was mounted upside down on the tip of the tubing and inserted into soapy water. The pressure inside the chamber was set through the machine console and checked by a second pressure probe. Pressure values started at 5 mmHg and were increased by 1 mmHg steps until the valve leaked air. Bubbling in the liquid soap on the valved trocar cannula made leakage apparent.

For the measurement of the pressure drop, the setup in Figure 4 was instead used: The infusion line was connected to the mock vitreous chamber by means of a three-way tap and, again, the pressure gauge was positioned in such a way as to measure the pressure of the chamber relative to the atmosphere. The membrane of the chamber was perforated with two valved trocar cannulas, the surface of which was wet with soapy water. The pressure inside the chamber was controlled by the Revolution CR 800 (BVI, Whaltam, MA, USA) console: 16 pressures in the range 5–60 mmHg were investigated.

As for the experiment using BSS, a test reference experiment using an unperforated membrane to close the mock chamber was performed. This allowed the tracing of a reference curve at zero leakage (Figure 5). For each brand and for each gauge of the trocars, two series of measurements were repeated.

## 3. Results

### 3.1. BSS Infusion Results

The opening pressure in BSS for all tested valved trocar cannulas is reported in Figure 6 and Table 1. The reported data show significant differences among brands, regardless of the considered gauge (*p* < 0.01 in all cases).

The average opening pressure was 7.69 ± 0.77 mmHg for 23G, 9.92 ± 0.57 mmHg for 25G and 7.57 ± 0.80 mmHg for 27G, and is reported in Figure 5b. The 25G valved trocar cannulas required a significantly higher (*p* < 0.05) BSS opening pressure than the other gauges, which did not differ significantly.

All valved cannulas started leaking at their respective opening pressure, as expected; the relation between piezometric bottle height and measured intraocular pressure is reported in Figure 7. The difference between expected and measured pressure within the mock vitreous chamber never exceeded 4 mmHg, regardless of gauge and/or manufacturer (see Figure 8).

Intraocular pressure loss due to valve leakage increased with bottle height and never exceeded 4 mmHg below the preset value, regardless of manufacturer and gauge. There was a trend towards less IOP reduction for the 27G cannulas compared to 25G and 23G, but it did not reach statistical significance. The difference between brands in IOP reduction due to leakage became significant for values greater than 9 mmHg at 23G, 25 mmHg at 25G and 30 mmHg at 27G (*p* < 0.05 in all cases). For IOP requests between 10 and 30 mmHg, no tested cannula lost more than 3 mmHg.

### 3.2. Air Infusion Results

The valve aperture pressure under air infusion varied significantly among the tested gauges (*p* < 0.01 at 23G and 27G, *p* < 0.05 at 27G; Figure 9). Overall, the 27G trocar cannulas showed a higher opening pressure than the 25G and 23G (*p* < 0.05), which behaved similarly.

We additionally compared the air pressure within the mock vitreous chamber with the console preset pressure. The reference manometer measured pressure completely corresponding to the machine preset values and there was no significant difference between the reference pressure and any used trocar cannula system, regardless of manufacturer and considered gauge.

## 4. Discussion

Ensuring volume and pressure stability during Pars Plana Vitrectomy (PPV) [9,10] is extremely important to maintain ocular homeostasis and prevent potentially blinding complications such as choroidal hemorrhage [11], ciliary effusion syndrome [12] and optic nerve damage [13].

Ever since the introduction of transconjunctival 25G vitrectomy by Fuji et al. [14] and 23G by Eckardt [15], the use of cannulas has been necessary to keep sclerotomy accessible and patent while leaving the conjunctiva virtually untouched. Cannulated sclerostomy systems offered the advantages of a faster recovery due to conjunctival integrity and a lower incidence of iatrogenic peripheral retinal tears [16], but at the same time exposed the eye to the risks related to intermittent hypotony upon instrument removal, especially when infusing air. The introduction of valves largely solved this problem but, to date, there is still a certain lack of data regarding the capability of valved cannulas to guarantee a safe intraocular pressure throughout surgery.

Valved cannula designs vary according to the manufacturer and can be divided into two different concepts: On the one hand, the single piece comprises removable valves made of rubber or silicone of variable thickness and stiffness, which are designed as plastic covers placed on the cannula’s proximal opening and retained by their own elasticity (Figure 1). A central crosshair cut makes four leaflets, allowing instruments’ introduction. On the other hand, there are non-removable valves embedded within the proximal aperture of the cannula, which are made of different leaflets that overlap.

Our study shows that all valves leaked over a certain “opening pressure”, reached at pressures between 3 and 16 mmHg in BSS (Figure 6) and 10–55 mmHg in air (Figure 9), and that BSS leakage rose as bottle height increased, explaining the divergence of data points in Figure 7 and Figure 8.

Pressure drops within the mock vitreous chamber never exceeded 4 mmHg and remained within 3 mmHg for bottle heights of between 10 and 30 mmHg, representing by far the most frequent IOP values requested during surgical procedures; however, statistically significant differences were shown among manufacturers (Figure 7 and Figure 8). The reason for such a reduced pressure drop despite leakage, often starting at low bottle heights, resides in the very limited flow generated by the leak, as delta pressure relates to energy loss, which is a function of flow rate.

Valve leakage and the resulting IOP drop do not seem to compromise safety, especially within the range of pressure usually set during surgery (15–35 mmHg). The difference between manufacturers, although statistically significant over a certain value, can in our opinion be considered “surgically negligible”, especially if we consider that all surgical maneuvers implying flow through instruments (vitreous cutter, extrusion needle, etc.) also determine IOP reduction, and minor deviations from the preset IOP continuously occur during the procedure.

A modest amount of leakage, especially at higher infusion pressures, can therefore be considered acceptable and even desirable, in that it mitigates the intraocular pressure when no instrument is in use. On the other hand, a completely watertight valve would certainly lead to higher friction along the instrument’s shaft, thus making surgical maneuvers more difficult and interfering with delicate movements such as membrane peeling. A tighter valve would also increase the risk of inadvertent valve removal from the cannula upon instrument retraction and removal, due to higher friction.

The aperture pressure varied between 10 and 50 mmHg under air infusion (Figure 9) and, on average, the 27G trocars started leaking at a higher pressure than the 25G and 23G. It is important to stress that valves maintain ocular pressure during the many downtimes of surgery, in between instrument changes, more than during surgical maneuvers since the presence of any instrument within the cannula virtually obstructs its lumen completely, due to minimal tolerance.

Valves are most needed under air infusion, since air viscosity is 2 orders of magnitude lower than BSS, and no pump or feedback mechanism can compensate for the exceedingly high flow and instantaneous drop in pressure occurring after instrument removal if the trocar cannula lumen is left open. Due to low viscosity, in fact, the air flow rates through patent trocar cannulas of any size are so high that the compensating ability of the pump is overwhelmed, and the eye pressure instantaneously drops to ambient pressure with variable globe collapse, with risks of bleeding, retinal dehydration damage [17] and even air embolism [18].

Despite leaking air, all tested valves successfully prevented ocular hypotension: even those with the lowest aperture pressure (around 10 mmHg, see Figure 8) proved “surgically” efficient, as they all displayed enough resistance to maintain the pump preset pressure almost completely.

Overall, the smaller gauges showed better air-sealing capacity than the larger ones, with a higher average opening pressure of 27G compared to 25G and 23G (Figure 8). The reason for this could be related to construction details such as the larger valve membrane required to cover the proximal aperture of wider trocars, longer crosshair cuts in larger gauges that make trocar cannulas slightly more susceptible to leakage, or possibly differences in valve material stiffness.

Different manufacturers chose two different design strategies based on different docking solutions to the infusion cannula (Figure 1): B+L, BVI, DORC and Optikon placed a removable silicon cap with crosshair cuts on top of the proximal aperture of the trocar cannula, whereas Alcon chose to embed a perforated membrane within the cap on the trocar cannula. Both solutions appeared similarly effective in reaching the desired pressure stability under both air and BSS infusion. However, the quest for the perfect surgical entry site is far from being fulfilled: perfectly watertight valves would still be highly desirable in that even minor leakage prompts vitreous traction and reduces vitreous cutters’ efficiency.

In summary, we tested the efficiency of leading manufacturers’ valved cannulas under BSS and air infusion and demonstrated that despite their leakage, they all maintained safe intraocular pressures within 4 mmHg of the desired value, with small surgically negligible differences. Pitfalls of the present study include the difficulty of measuring very minute amounts of leakage and therefore obtaining exact measurements of the “aperture pressure”, in BSS more than under air infusion, and the need to use a single air pump from a given manufacturer to test air leakage to reduce variability. On the other hand, the high number of measurements allowed statistical significance to be obtained.

## Figures and Tables

**Figure 1 bioengineering-12-00431-f001:**
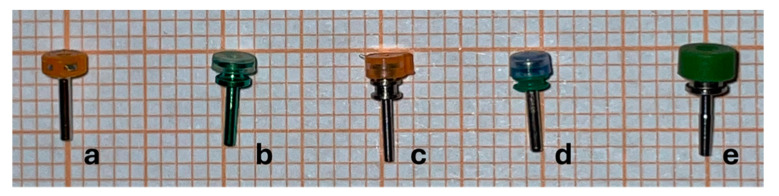
Valved trocar cannulas tested (25G series): (**a**) Alcon; (**b**) Bausch & Lomb; (**c**) BVI; (**d**) DORC; (**e**) Optikon 2000.

**Figure 2 bioengineering-12-00431-f002:**
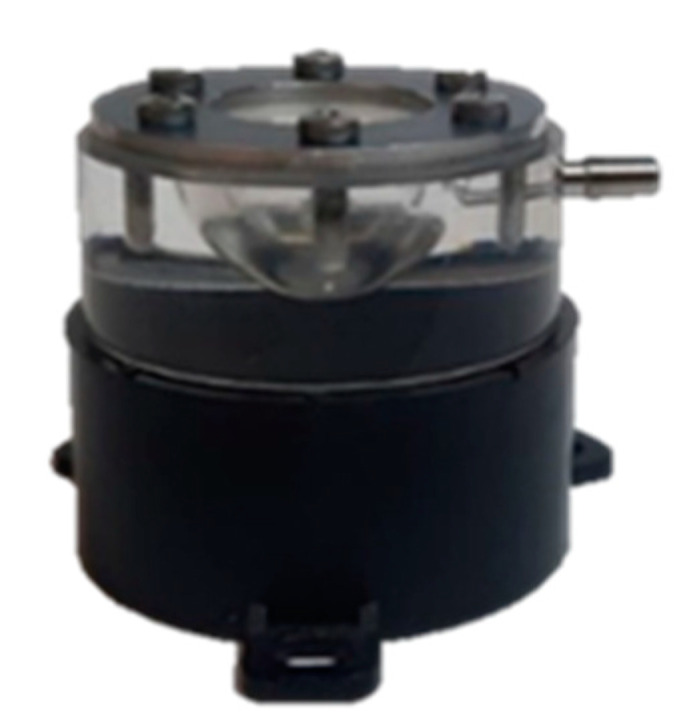
Mock vitreous chamber: a hollow hemisphere made of Plexiglas with 4 cc volume to simulate the vitreous chamber, closed at the top by a silicone membrane allowing trocar puncture and watertight cannula placement for infusion.

**Figure 3 bioengineering-12-00431-f003:**
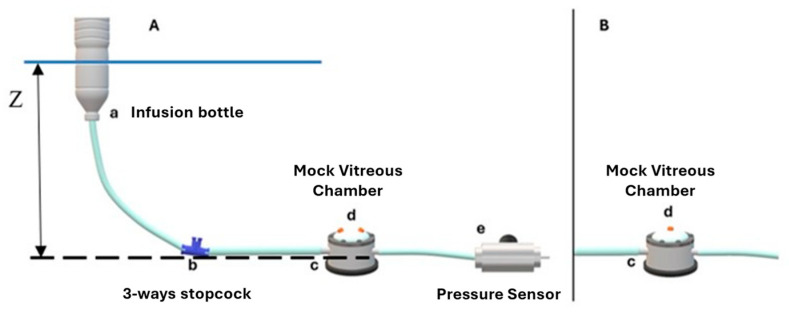
(**A**) Sketch of the setup for pressure–pressure curve measurements: (a) open bottle; (b) 3-way stopcock with the empty part closed; (c) vitreous chamber simulator; (d) double-valved cannula layout for pressure loss measurements; (e) pressure probe; all elements are connected with rubber tubes (internal diameter of 4 mm). (**B**) Detail of mock vitreous chamber (c) with single-valved cannula layout (d) used to measure opening pressure of the valve.

**Figure 4 bioengineering-12-00431-f004:**
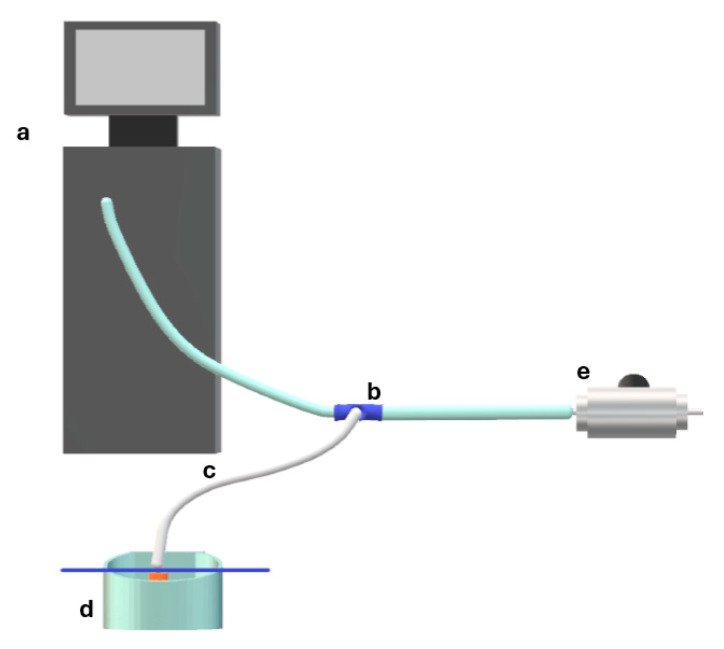
Setup used for measuring the leakage pressure of trocar cannula valves in air: (a) R-evolution CR vitrectomy console; (b) 3-way stopcock; (c) tubing and trocar cannula; (d) tray filled with soapy water to the level indicated by the blue line; (e) pressure probe.

**Figure 5 bioengineering-12-00431-f005:**
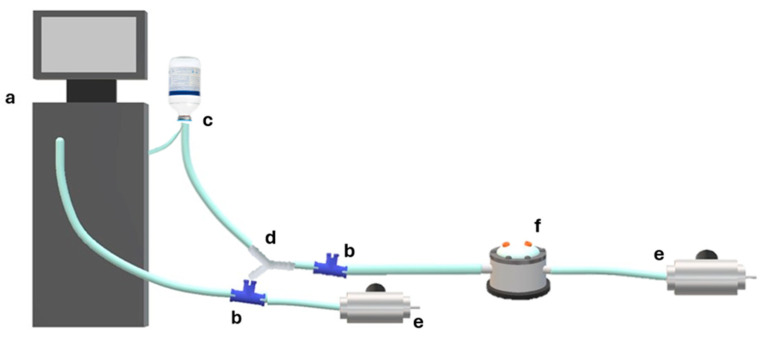
Setup used to construct the reference air pressure–pressure curves: (a) R-evolution CR console machine; (b) 3-way stopcock; (c) pressurized bowl; (d) Y-junction; (e) differential pressure sensor; (f) mock vitreous chamber and trocar cannula layout.

**Figure 6 bioengineering-12-00431-f006:**
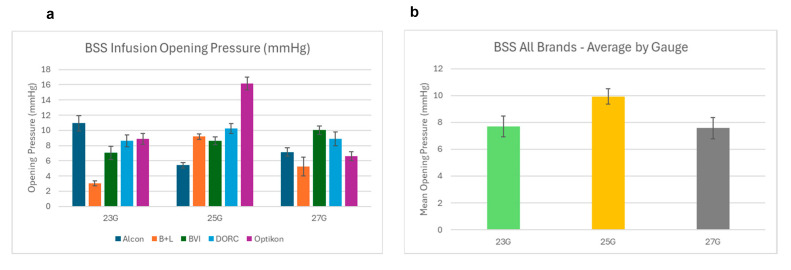
(**a**) BSS opening pressure (in mmHg) for all tested valved trocar cannulas. (**b**) Average opening pressure (in mmHg) by gauge.

**Figure 7 bioengineering-12-00431-f007:**
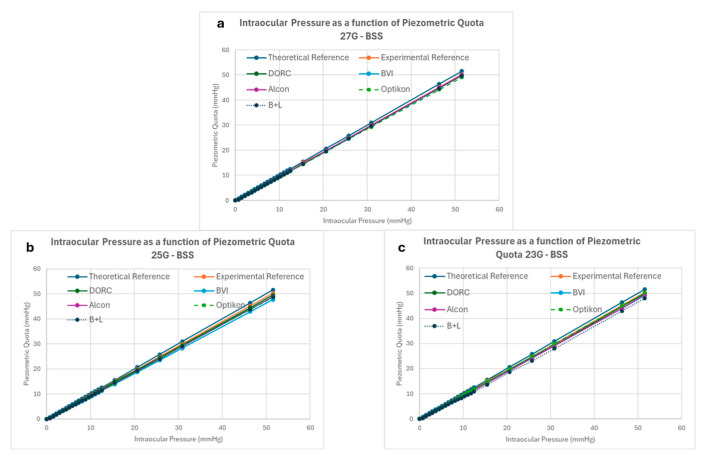
Intraocular pressure as a function of BSS infusion bottle height (mmHg): (**a**) 23G: (**b**) 25G; (**c**) 27G.

**Figure 8 bioengineering-12-00431-f008:**
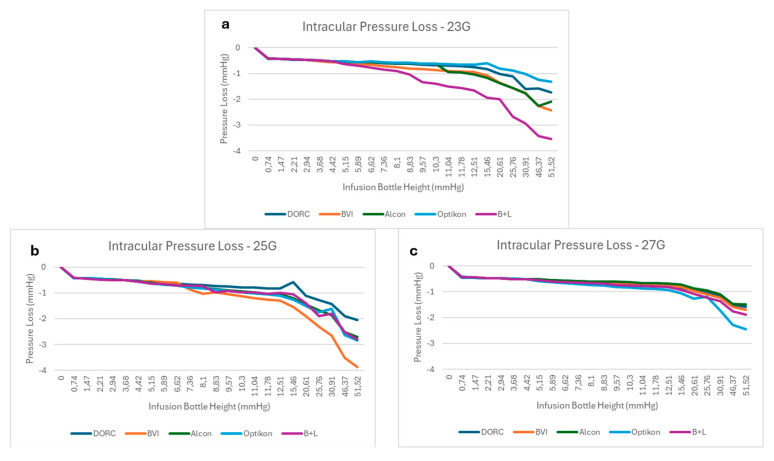
Pressure loss by gauge and manufacturer, as a function of BSS infusion bottle height (in mmHg).

**Figure 9 bioengineering-12-00431-f009:**
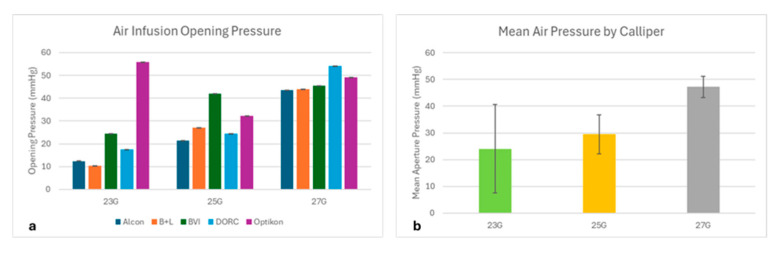
Mean aperture pressure valve under air infusion divided per (**a**) brand and (**b**) gauge.

**Table 1 bioengineering-12-00431-t001:** Valved cannulas’ opening pressure under BSS infusion (in mmHg).

	23G	25G	27G
Alcon	10.94 ± 1	5.42 ± 0.36	7.13 ± 0.55
B+L	3.02 ± 0.34	9.18 ± 0.35	5.25 ± 1.23
BVI	7.03 + 0.87	8.62 ± 0.52	10.02 ± 0.54
DORC	8.6 ± 0.78	10.25 ± 0.65	8.87 ± 0.9
Optikon	8.88 ± 0.71	16.15 ± 0.84	6.58 ± 0.59
*p* value	<0.01	<0.01	<0.01

## Data Availability

All raw data are available upon request to the corresponding author.

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
