# Peer review of "Comparing the Efficiency of Valved Trocar Cannulas for Pars Plana Vitrectomy"

_bioengineering, 2025, doi:10.3390/bioengineering12040431_

Round 1

Reviewer 1 Report

Comments and Suggestions for Authors

The present manuscript experimentally compares and verifies the application parameters of valved trocar cannulas from five different manufacturers with different diameters (gauges) for effective vitrectomy surgery. The objectives of the work are clearly defined and understandable; on the other hand, the question remains as to what extent this is a research activity that does not result in a new finding, a new solution, or even a significantly new insight. It is also unclear what raised doubts about the correct functional parameters of the cannulas studied, whether any failures occurred or are occurring during the eye surgeries.  In other words, what problem are the authors targeting, and how do they propose to solve it based on their findings? These aspects should be clarified. In the experimental section, the authors appropriately focus on obtaining the necessary parameters and thus purposely conduct several main experiments that are sufficient to compare and obtain the necessary data. The analysis of the experimental data is admittedly clear and sufficiently detailed. However, there are doubts about the correct methodology. Firstly, a threshold of "surgical significance" is identified, which is not sufficiently cited, and it is not clear where exactly it comes from. At the same time, it seems relatively high, benevolent compared to the measured results. How are the "best-in-class and worst-in-class" products related to this threshold? What does this mean in particular?  What can then be considered an appropriate and undesirable result, since the authors end up rating no product significantly either bad or good. The text should be supplemented with values or ranges of monitored parameters that are essential for the correct function of the products during eye surgery. This is the basis for assessing individual products.   Second, in terms of methodology, it is important to average the observed parameters across products from different suppliers, but it will be essential instead to highlight and focus on their differences. Why do products for the same application exhibit different behaviour and whether this is ok against "surgical significance". In the first results, e.g. Fig6A, we see high differences, but they average out and disappear in FIG6B. This should be handled differently.   Because of these shortcomings, I cannot recommend the manuscript for publication.   I add a few more comments:

  • BSS is not explained when first used.
  • Missing keywords.
  • Fig.2 is not descriptive enough. A diagram indicating the essential parameters would be much more welcome.
  • FIG6B states in the caption the word "calliper", is it the correct caliber with gauge units? See line 164 as well.
  • The sentence in lines 142-145 is confusing; check the references to the figures.
  • The labels of the graphs in Figure 7 are difficult to read.
  • The graph in Figure 8 requires a more detailed description of how each point was obtained.
  • The PPV on line 166 is not explained.
  • In the sentence on lines 171, 199, it should be patient instead of patent.
  • The conclusion is missing, with a summary of the results, their application, and any suggestions for further research.

Author Response

The present manuscript experimentally compares and verifies the application parameters of valved trocar cannulas from five different manufacturers with different diameters (gauges) for effective vitrectomy surgery. The objectives of the work are clearly defined and understandable; on the other hand, the question remains as to what extent this is a research activity that does not result in a new finding, a new solution, or even a significantly new insight. It is also unclear what raised doubts about the correct functional parameters of the cannulas studied, whether any failures occurred or are occurring during the eye surgeries.  In other words, what problem are the authors targeting, and how do they propose to solve it based on their findings? These aspects should be clarified. In the experimental section, the authors appropriately focus on obtaining the necessary parameters and thus purposely conduct several main experiments that are sufficient to compare and obtain the necessary data. The analysis of the experimental data is admittedly clear and sufficiently detailed. However, there are doubts about the correct methodology. Firstly, a threshold of "surgical significance" is identified, which is not sufficiently cited, and it is not clear where exactly it comes from. At the same time, it seems relatively high, benevolent compared to the measured results. How are the "best-in-class and worst-in-class" products related to this threshold? What does this mean in particular?  What can then be considered an appropriate and undesirable result, since the authors end up rating no product significantly either bad or good. The text should be supplemented with values or ranges of monitored parameters that are essential for the correct function of the products during eye surgery. This is the basis for assessing individual products.   Second, in terms of methodology, it is important to average the observed parameters across products from different suppliers, but it will be essential instead to highlight and focus on their differences. Why do products for the same application exhibit different behaviour and whether this is ok against "surgical significance". In the first results, e.g. Fig6A, we see high differences, but they average out and disappear in FIG6B. This should be handled differently.   Because of these shortcomings, I cannot recommend the manuscript for publication.  

First of all, we would like to thank the reviewers for the time spent on our manuscript and their precious counseling that prompted a significant improvement of it

The concept underlying the entire manuscript is to measure the amount of leakage and more importantly, measure the pressure drop related to flow 

I add a few more comments:

  • BSS is not explained when first used.

Corrected

  • Missing keywords.

Keywords are actually present on line 30, right after the abstract.

  • Fig.2 is not descriptive enough. A diagram indicating the essential parameters would be much more welcome.

The mock vitreous chamber has been described into details in fig. 2 caption and we hope it is clearer now.

  • FIG6B states in the caption the word "calliper", is it the correct caliber with gauge units? See line 164 as well.

Corrected, (also in fig. 8)

  • The sentence in lines 142-145 is confusing; check the references to the figures.

The sentence has been thoroughly rephrased and is hopefully now clearer.

  • The labels of the graphs in Figure 7 are difficult to read.

The figure 7 has been re-done and font increased to 12 points

  • The graph in Figure 8 requires a more detailed description of how each point was obtained.

The caption of figure 8 has been improved and is now clearer

  • The PPV on line 166 is not explained.

Corrected

  • In the sentence on lines 171, 199, it should be patient instead of patent.

"Patent" is actually correct, meaning in this case "pervious aperture", a hole that allows flow through it

  • The conclusion is missing, with a summary of the results, their application, and any suggestions for further research.

the conclusion has been widely re-formulated and includes those suggestions now

We herein include the revised manuscript with track changes on, while the "clean" new version is attached on the main page. Thank you again. Tommaso Rossi MD

Reviewer 2 Report

Comments and Suggestions for Authors

Thank you for the opportunity to review this manuscript comparing the efficiency of different manufacturers' valved cannulas for pars plana vitrectomy. This study provides a methodical analysis of an important aspect of vitreoretinal surgery that is rarely evaluated objectively.

The study addresses a practical surgical issue that is relevant to everyday vitreoretinal practice but has not been well-characterized in the literature. The methodology is clearly outlined with appropriate experimental designs for evaluating both BSS and air leakage; moreover, the comparison across multiple manufacturers and three gauge sizes (23G, 25G, 27G) is comprehensive and of clinical value.

Areas for Improvement

1. Introduction
- Please add prior studies examining cannula valve function 

2. Methods
- For the BSS infusion setup: please specify the exact height increments used between 0.10 m and 0.27 m, as well as between 0.27 m and 0.80 m.
- The study requires a justification for the choice of 4 mmHg as the threshold for "surgical significance" in pressure difference.
- For Figure 3, please add labels directly to the components in the figure to improve clarity.
- Please clarify whether the same instruments were used across all tests for consistency (e.g., same vitrectomy cutters or forceps).

3. Results
- Figure 6a is somewhat difficult to interpret due to the multiple bars and colors. Please revise the different visualization or adding a table with the specific values.
- Please include the raw data for the opening pressures of each brand and gauge; this can be done as a supplementary table.
- For Figure 7, revise using different line styles (dotted, dashed, etc.) in addition to colors to improve accessibility.
- Include statistical analysis (p-values) for the differences between manufacturers within each gauge size.

4. Discussion
- The discussion requires addressing the clinical implications of the differences found between manufacturers; would these differences impact surgical decision-making?
- Please discuss the potential implications for specific surgical scenarios where pressure stability is particularly critical (e.g., silicone oil removal, diabetic tractional detachment surgery).
- Please contribute any limitations in the experimental setup that might not perfectly replicate in vivo conditions.

RECOMMENDATIONS: This manuscript presents utility on the performance of different valved trocar cannulas that will be of interest to vitreoretinal surgeons. With the revisions above to improve clarity and address shortcomings, this paper can make a useful contribution to the literature. I recommend revise to address these suggestions; I would be happy to review an update.

Author Response

First of all we would like to thank the reviewers for the time spent on our manuscript and their precious counseling that prompted a significant improvement of it

Introduction

- Please add prior studies examining cannula valve function

Done

2. Methods
- For the BSS infusion setup: please specify the exact height increments used between 0.10 m and 0.27 m, as well as between 0.27 m and 0.80 m.

that has been duly described between lines 104-113: 10mm steps between 7-20mmHg and 70mm (5mmHg) after that.

  • The study requires a justification for the choice of 4 mmHg as the threshold for "surgical significance" in pressure difference.

Indeed that concept was badly exposed: we completely re-phrased all occurrences of that concept. The loss of intaocular pressure  due to valve leakage never exceeded 4 mmHg regardless of manufacturer and considered gauge, even when the preset pressure was over 50 mmHg. In the new version we comment within the the discussion section that this is probably negligible from a surgical standpoint.

  • - For Figure 3, please add labels directly to the components in the figure to improve clarity.
    - Please clarify whether the same instruments were used across all tests for consistency (e.g., same vitrectomy cutters or forceps).

The figure 3 has been modified introducing callout for each piece of the setting

3. Results
- Figure 6a is somewhat difficult to interpret due to the multiple bars and colors. Please revise the different visualization or adding a table with the specific values.
- Please include the raw data for the opening pressures of each brand and gauge; this can be done as a supplementary table.

a table with mean and standard deviation of aperture pressure has been introduces with p values for each brand

- For Figure 7, revise using different line styles (dotted, dashed, etc.) in addition to colors to improve accessibility.

Line styles for different series have been changed but the graph remains fairly crowded as series almost overlap. We therefore decided to change completely fig. 8 and reports as pressure delta the difference between desired and actual pressure for each gauge and manufacturer. Since the difference becomes more evident as requested pressure increases, lines diverge more evidently and we believe this offers a better view of data. 

- Include statistical analysis (p-values) for the differences between manufacturers within each gauge size.

Since this requirement would imply a very busy table for each test point from 0-51 mmHg, we included in the results section the value over which data divergence reached statistical significance.

4. Discussion
- The discussion requires addressing the clinical implications of the differences found between manufacturers; would these differences impact surgical decision-making?

That has been introduced

- Please discuss the potential implications for specific surgical scenarios where pressure stability is particularly critical (e.g., silicone oil removal, diabetic tractional detachment surgery).
- Please contribute any limitations in the experimental setup that might not perfectly replicate in vivo conditions.

The discussion has been largely modified, according to those suggestions

RECOMMENDATIONS: This manuscript presents utility on the performance of different valved trocar cannulas that will be of interest to vitreoretinal surgeons. With the revisions above to improve clarity and address shortcomings, this paper can make a useful contribution to the literature. I recommend revise to address these suggestions; I would be happy to review an update.

We herein include the revised manuscript with track changes on, while the "clean" new version is attached on the main page. Thank you again. Tommaso Rossi MD

Round 2

Reviewer 1 Report

Comments and Suggestions for Authors

I would like to thank the authors for their efforts in editing the article based on my comments. The abbreviation BSS remained in the abstract without explanation. Otherwise, I have no further comments and agree with the publication.

Reviewer 2 Report

Comments and Suggestions for Authors

The authors have reviewed & approved appropriately, including: 

  • Keywords: The keywords have been updated to include "Pars Plana Vitrectomy," "Flow Rate," "Head Loss," and "Valved Trocar Cannulas".
  • Introduction: The introduction has been revised to clarify the purpose of the surgery and the importance of maintaining invariant pressure and volume in the vitreous chamber. It also includes a sentence about previous studies on valved cannulas. The last paragraph of the introduction has been significantly rewritten to emphasize the lack of comparative data on valved cannulas and clearly state the study's purpose.
  • Materials and Methods - Figure 2: The description of Figure 2 has been updated to provide more detail about the mock vitreous chamber.
  • Materials and Methods - Figure 3: The description of Figure 3 has been revised for clarity.
  • Materials and Methods—Valve opening pressure assessment: The method for assessing valve opening pressure has been slightly rephrased for better clarity.
  • Materials and Methods - Air infusion set-up: The description of Figure 4 has been updated.
  • Materials and Methods—Air reference pressure-pressure curves: The method for constructing air reference pressure-pressure curves has been revised for clarity.
  • Results—BSS infusion Results: Table 1 has been added to present the valved cannula opening pressure under BSS infusion. The text referring to Figure 6 has been updated to include a reference to Table 1.
  • Results - Figure 7: The description of Figure 7 has been updated.
  • Results - Figure 8: The description of Figure 8 has been updated to clarify that the pressure loss is shown by gauge and manufacturer. The text has been expanded to provide more details on the pressure loss at different infusion pressures and to mention the statistical significance of the differences between brands.
  • Results - Air Infusion Results: The text has been revised to state that the valve aperture pressure varied significantly among tested gauges.
  • Results - Figure 9: The description of Figure 9 has been updated.
  • Discussion: The discussion section has been significantly expanded and rewritten. It now includes more details about the clinical implications of the findings, the design variations among different manufacturers' valves, and the limitations of the study and future directions.
  • References: Several new references have been added.

The second version is significantly improved based on these revisions, and the revised manuscript merits publication.